# How the motor aspect of speaking influences the blink rate

**Mareike Brych**[ID]*, **Supriya Murali, Barbara Händel**

Department of Psychology III, University of Würzburg, Würzburg, Germany

* mareike.brych@uni-wuerzburg.de

**Data Availability Statement:** The data is available on OpenScienceFramework (http://doi.org/10.17605/OSF.IO/JT3V7).

**Funding:** This study was supported by a starting grant from the European Research Council awarded to B. Händel (grant number 677819;

## Abstract

The blink rate increases if a person indulges in a conversation compared to quiet rest. Since various factors were suggested to explain this increase, the present series of studies tested the influence of different motor activities, cognitive processes and auditory input on the blink behavior but at the same time minimized visual stimulation as well as social influences. Our results suggest that neither cognitive demands without verbalization, nor isolated lip, jaw or tongue movements, nor auditory input during vocalization or listening influence our blinking behavior. In three experiments, we provide evidence that complex facial movements during unvoiced speaking are the driving factors that increase blinking. If the complexity of the motor output increased such as during the verbalization of speech, the blink rate rose even more. Similarly, complex facial movements without cognitive demands, such as sucking on a lollipop, increased the blink rate. Such purely motor-related influences on blinking advise caution particularly when using blink rates assessed during patient interviews as a neurological indicator.

## Introduction

Humans blink approximately every 3–6 seconds, which is far more than needed to keep a constant tear film on the cornea [1]. Blink behavior is known to be affected by multiple factors, including external sensory [e.g., 2,3] and internal cognitive factors [e.g., 4,5]. During visually demanding tasks such as reading, the blink rate drops from approximately 17 blinks per minute during rest to approximately 4 to 5 blinks per minute. Conversely, the blink rate increases during conversation to approximately 26 blinks per minute [6]. This increase has been proposed to reflect various internal processes such as engagement, emotions or opinions [7]. Hömke, Holler and Levinson [8] further showed that blinks can serve as communicative signals between conversation partners. Findings as to the role of motor execution on blink rate are inconsistent. Research has shown that the motor act of speaking [9], but not jaw movements as produced during gum chewing [4] or the mere act of keeping the mouth open [10] increased blinking. Interestingly, in the latter study, a small group that exhibited notable mouth and jaw movements during a no-task condition nearly had a doubled blink rate compared to those who did not show such movements. A clarification of the influence of motor activity seems relevant, especially since blinks serve as neurological indicators in clinical

https://erc.europa.eu/). The funders had no role in study design, data collection and analysis, decision to publish, or preparation of the manuscript.

**Competing interests:** The authors have declared that no competing interests exist.

settings. For example, very low blink rates are observed in patients with Parkinson's disease [11], which is possibly due to dopaminergic hypoactivity [review by 12, questioned by 13,14]. The patient's response to medication can be assessed by the increase in blink rate, which is often measured during the conversation with the physician [11]. Consequently, if other factors such as speaking increases blink rate in the same direction, this might lead to inaccurate medical examinations. In healthy humans, blink rate is often used as an indicator of cognitive load [e.g., 4,5,15]. A speech-related motor influence might therefore affect experimental outcomes using verbal responses.

We set up an experiment to systematically investigate the influences of facial motor activity on blinking behavior, while at the same time controlled cognitive and auditory influences. Several anatomical findings reveal that the eyelid and facial muscles are connected. Speaking involves various motor processes including the respiratory system, larynx and vocal tract, which is shaped by the lips, jaw and tongue [16]. In the human brain, the area for vocalization is located inferior to the area for eyelid movements and superior to the areas for mouth movements including tongue and lip movements. The area for jaw movements is inferior to the mentioned mouth movements [17]. Considering human facial anatomy, the facial nerve (7th cranial nerve) innervates the muscles for facial expressions and eyelid closing, but is not directly involved in chewing movements [18]. Whenever the facial nerve malfunctions, blinking is ceased and the corner of the mouth drops on the affected side [19]. During surgeries, facial nerve stimulation is also used to predict the postoperative function by checking motor-evoked potential in the eye ring muscle (orbicularis oculi) and the kissing muscle (orbicularis oris) [20]. The above reviewed work clearly shows a proximity of the anatomical substrate of blinking and other facial movements. Our experiments particularly test the influence of motor activity on the blink rate including the isolated movements of the lips, jaw and tongue as well as speech-related movements with and without vocalization. Apart from the new insights on how blinks and other body movements are related, our work seeks to clarify the validity of blinks as a marker for pathological states as well as for sensory and cognitive processing in experiments using verbal responses.

## Experiment 1

In a first experiment, we tested for influences of motor output during speaking. In order to account for the auditory and cognitive aspect, we included conditions in which we varied the cognitive as well as the auditory input normally introduced by speaking. We hypothesize that the blink rate is mainly increased by motor related factors as indicated by the proximity of anatomical conditions [17,18] as well as by previous research concluding a motor effect, but without strict control of other possible influences [9]. Only few studies investigated blink behavior under auditory stimulation. Concerning the number of blinks during a task, these studies reported no significant changes compared to rest [2,21] and studies testing for cognitive influences are inconsistent [22,23]. Therefore, we assume that auditory input or cognitive aspects of speaking only have a minor effect on blinking. Visual stimulation as well as social influence were minimized in our experiment.

### Method

**Participants.**   30 psychology students of the University of Würzburg (mean age: 20.17 years ± 1.86 SD, 2 male) took part in the study. All participants gave their written informed consent and received study credit for their participation. The study was approved by the local ethics committee (Institute for Psychology of the Faculty for Human Sciences of the Julius-

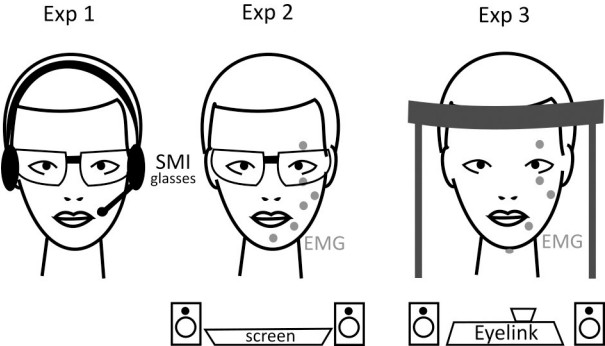

**Fig 1. Experimental setup for the three experiments.** In experiment 1, we recorded eye movements with SMI eye tracking glasses. In experiment 2, we added EMG and in experiment 3, we used an Eyelink eye tracker and EMG.

Maximilians-University of Würzburg; project protocol number: GZEK 2015–01) and was in line with the European general data protection regulations (DSVGO).

**Procedure.** Participants sat alone in a noise shielded, very small, dimly lit room. They were allowed to freely move their eyes and head. Auditory instructions were given by a Sennheiser PC3 Chat headset. Binocular eye movements were recorded with the 120Hz SMI eye tracking glasses (Fig 1).

When measuring blink rate during a conversation, there are several possible influences. Our different experimental conditions were designed to test for influences of the cognitive load during speech production (with and without vocalization), of motor output (mouth movements, with focus on lip or jaw movements) and of auditory input (due to one's own speaking or someone else). The study consisted of eight different tasks, which were repeated 5 times (except for the baseline, which was repeated 15 times) and each lasted for 1 minute. The tasks were "normal talking", "talking inside the head", "talking without sound", "lip movement", "jaw movement", "listen to someone else", "listen to oneself" and "baseline" (being at rest). Table 1 summarizes all tasks. During "normal talking", "talking inside the head"and "talking without sound", participants were instructed to talk about easy topics like "Describe your apartment"or "Describe your last holiday". Topics were defined by us and randomized across tasks and participants. "Talking inside the head"involved no mouth movement and no sound production, but required cognitive processes that are comparable to the cognitive

**Table 1. List of tasks, their description and their use in the analysis.**

| Task | Description | Analysis of which effect |
|---|---|---|
| **"normal talking"** | Talk about a given topic **with** mouth movements and **with** vocalization | Cognitive (Fig 2) |
| **"talking inside the head"** | Talk about a given topic **without** mouth movements and **without** vocalization | Cognitive (Fig 2) |
| **"talking without sound"** | Talk about a given topic **with** mouth movements, but **without** vocalization | Motor (Fig 3) |
| **"lollipop"** | Sucking on a lollipop to induce lip movement | Motor (Fig 3) |
| **"gum"** | Chewing a gum to induce jaw movement | Motor (Fig 3) |
| **"listen to someone else"** | Listen to an unknown monologue of a woman | Auditory (Fig 4) |
| **"listen to oneself"** | Listen to own monologue recorded during "normal talking" | Auditory (Fig 4) |
| **"baseline 1–3"** | Resting | All (Figs 2–4) |

processes during "normal talking". „Talking without sound"referred to simply mouthing words mimicking mouth movements during "normal talking" but omitting auditory stimulation. To induce lip movements independent of talking, participants were asked to suck on a real lollipop ("lollipop"). In another condition ("gum"), chewing a gum resulted in jaw movements. We chose sucking on a lollipop as an easy way to induce mouth and especially lip movements. Respectively, gum chewing intended to mainly introduce jaw movements. However, we are aware that also other movements such as tongue movements and swallowing are likely executed as well. In the auditory conditions, auditory input was either a monologue of a young woman ("listen to someone else") or a playback of their own monologue recorded from a previous "normal talking"trial ("listen to oneself"). "Listen to oneself" therefore is the same auditory input as during the "normal talking" condition, however, "listen to someone else" was added to mimic the auditory input experienced during a conversation with another person. During the baseline conditions, participants should not stand up or close their eyes, but had no additional task, which will be referred to as 'resting'. "Baseline 1" consisted of 5 randomly selected minutes of the 15 baseline minutes, "baseline 2" of 5 randomly selected minutes of the 10 remaining minutes and "baseline 3" of the lastly 5 remaining minutes. This was done to prevent multiple testing of the same data. The order of tasks was fully randomized to exclude any time related effects, except that the task "listen to oneself"needed to be placed after the "normal talking"condition. Participants were able to start each trial at their own pace by pressing a button followed by a starting tone. The end of the trial was signaled by another tone.

**Data analysis.** Four participants were excluded (three due to more than 20% eye data loss, one due to an extremely high mean blink rate >50 blinks/min). Additionally, the eye recording of one participants was lacking two trials. The blink rates over the five repetitions of each task were averaged before comparing between tasks. Since we did not have the participant's permission to listen to the monologues, we plotted the recorded sound signal and visually inspected the amplitude of the signal representing speech to control for task fulfillment in the "normal talking" condition. Repeated measures ANOVAs and corresponding post-hoc analyses for blink rate were computed. The epsilon for Huynh-Feldt correction is given in case of violation of sphericity. Bayesian analysis was added as a supplement to the classical frequentist statistics to get insights on the credibility of the alternative as well as the null hypotheses. The experimental program was implemented and analyzed in MATLAB R2015b (Mathworks). Bayesian analysis was performed with JASP (JASP Team (2019), Version 0.11.1.0).

**Blink detection.** When the eyelid occludes the pupil during a blink, pupil size recordings of video-based eye tracker quickly and strongly decrease until the pupil is undetectable. Using this characteristic, our blink detection algorithm is based on the recorded pupil size. Blinks were initially detected when both z-transformed pupil radii were below a threshold of -2 standard deviations or when the pupil data was marked as lost. The start and the end of the blink were then shifted to the time point when the radii were higher than half the threshold. Blinks less than 50ms apart from each other were concatenated. Blinks longer than 1000ms and shorter than 50ms were discarded.

## Results Experiment 1

To test for cognitive influences on the blink rate, we compared "baseline 1" (no task) with "talking inside the head" (only the cognitive component of speaking) and with "normal talking". A repeated measures 1-factor ANOVA compared the blink rate between these tasks and revealed a significant main effect ($F(2,50) = 25.22$, $p < .001$, $\eta_p^2 = .502$, $\varepsilon = .679$, Huynh-Feldt correction (HF)). Post-hoc pairwise t-tests revealed a significant higher blink rate during

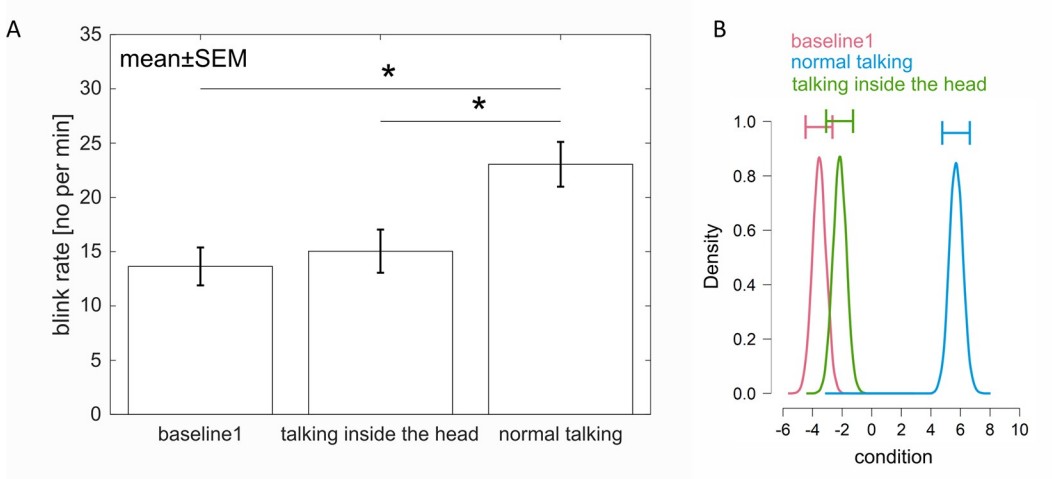

**Fig 2. Influence of the cognitive component on the blink rate.** A. Blink rate during "baseline1" (being at rest), "talking inside the head" and "normal talking". Error bars represent one standard error of the mean (SEM). Stars mark significant differences revealed by parametric statistics. B. Posterior distributions of the effect of each condition on the blink rate. "Normal talking" has highest effect on blink rate followed by "talking inside the head" and "baseline1". The horizontal error bars above each density represent 95% credible intervals.

"normal talking" than during "talking inside the head" ($p < .001$) as well as a significantly higher blink rate during "normal talking" than during "baseline 1"($p < .001$) (Fig 2A).

In addition to the classical Frequentist analysis, a Bayesian analysis was performed to improve possible interpretations of the results. Comparing the model with the predictor, that the tasks ("baseline 1", "normal talking" and "talking inside the head") have an effect on the blink rate, to the null model, overwhelming evidence for the alternative was revealed (Bayes Factor: $BF_{10} = 3.636^*10^5$). Post-hoc tests showed strong evidence that the blink rate during "normal talking" differed to the blink rate during "baseline 1" as well as to the blink rate during "talking inside the head" (adjusted posterior odds of $2.507^*10^3$ and $1.818^*10^2$). Additionally, there was evidence that the blink rate during "baseline 1" and "talking inside the head" were the same (adjusted posterior odds of 1/0.529 = 1.890) (Fig 2B).

In a next step, the influence of different motor components on the blink rate was investigated. Fig 3A shows a high blink rate during "talking without sound", followed by lip movements during "lollipop" sucking and jaw movements during "gum" chewing. The "baseline 2" condition with no movement showed the lowest blink rate. A repeated measures ANOVA showed a significant main effect of tasks on blink rate ($F(3,75) = 8.94$, $p < .001$, $\eta_p^2 = .263$, $\varepsilon = .800$ (HF)). Post-hoc tests specified this effect. The blink rate was significantly lower during the "baseline 2" compared to "lollipop" ($p = .016$) and compared to "talking without sound" ($p = .003$). Neither did the difference between "gum" chewing and "baseline 2" reach significance ($p = .106$), nor did any other comparison between movements ($ps > .105$).

Again, Bayesian ANOVA was additionally conducted to assess the differences in blink rate between tasks. Given the predictor of tasks ("baseline 2", "lolli", "gum" and "talking without sound"), strong evidence for the alternative was found when comparing the model with the predictor to the null model (Bayes Factor: $BF_{10} = 5.371^*10^2$). Post-hoc comparisons revealed moderate evidence for differences in blink rate between "baseline 2" and "lollipop" as well as between "baseline 2" and "talking without sound" (adjusted posterior odds of 6.086 and 29.963). The evidence for differences in blink rate between "baseline 2"and "gum" as well as between "gum" and "talking without sound" was rather inconclusive (odds of 1.212 and

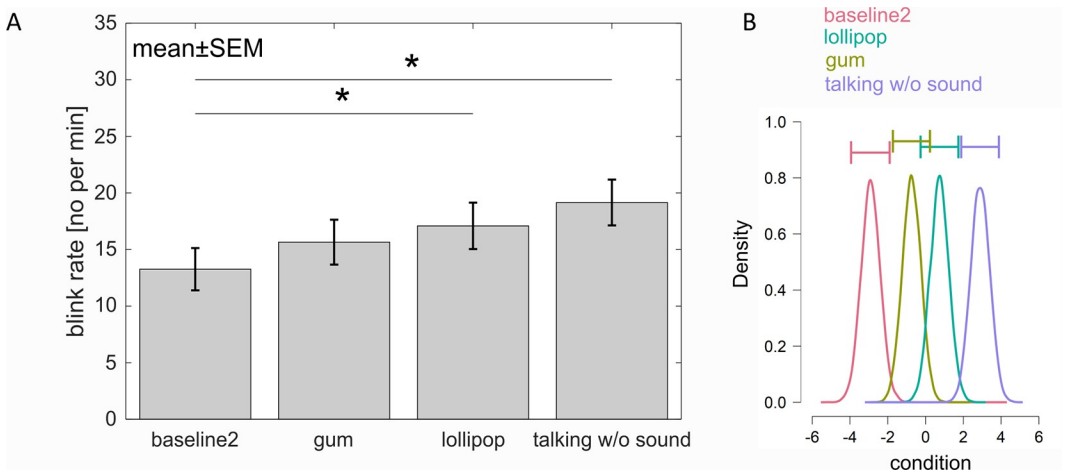

**Fig 3. Influence of motor tasks on the blink rate.** A. Blink rate during the second baseline (being at rest), moving the lips during lollipop sucking, moving jaw muscles during gum chewing and talking without sound production. Error bars represent one SEM. Stars mark significant differences revealed by parametrical statistics. B. Posterior distributions of the effect of each condition on the blink rate. Talking without sound has highest effect on blink rate followed by lip movement during lollipop sucking, jaw movement during gum chewing and baseline. The horizontal error bars above each density represent 95% credible intervals.

1.222). Blink rate between "gum" and "lollipop" as well as between "lollipop" and "talking without sound" was not different from each other (odds of 1/0.609 = 1.642 and 1/0.278 = 3.597) (Fig 3B).

Finally, the influence of auditory input on blink rate was examined with a repeated measures ANOVA comparing the blink rate between the conditions "baseline 3", "listen to oneself" and "listen someone else". The main effect suggesting a difference between conditions was significant ($F(2,50) = 3.96$, $p = .036$, $\eta_p^2 = .137$, $\varepsilon = .790$ (HF)). Post-hoc tests did not reveal a difference in blink rate between the "baseline 3" condition and any auditory input ($ps > .089$), but a significant difference between "listen to oneself" and "listen to someone else" ($p = .027$) (Fig 4A).

Bayesian analysis revealed evidence that the model with the predictor of tasks on the outcome of the blink rate is better than the null model ($BF_{10} = 2.022$). Post-hoc tests revealed that the blink rates between "baseline 3" and "listen to oneself" are not different from each other (1/0.163 = 6.135), while the blink rates between "listen to someone else" and "listen to oneself" are different (odds of 2.992). The data does not seem to be sufficiently informative to show whether there is a difference between "baseline 3" and "listen to someone else" or not (odds of 1.127) (Fig 4B).

## Discussion Experiment 1

Our results replicated previous findings that talking is accompanied by an increase in blink rate compared to baseline [e.g., 4]. More specifically, our results suggest that neither the cognitive processes nor the auditory input, but rather, the motor activity of the mouth has the main influence on our blink rate.

The conditions "talking inside the head" and "normal talking" differed in terms of motor output and auditory input but not cognitive processes, which are needed for the production of meaningful sentences. Since the blink rate was significantly lower during "talking inside the head" than during "normal talking" and highly similar to "baseline 1", cognitive processes

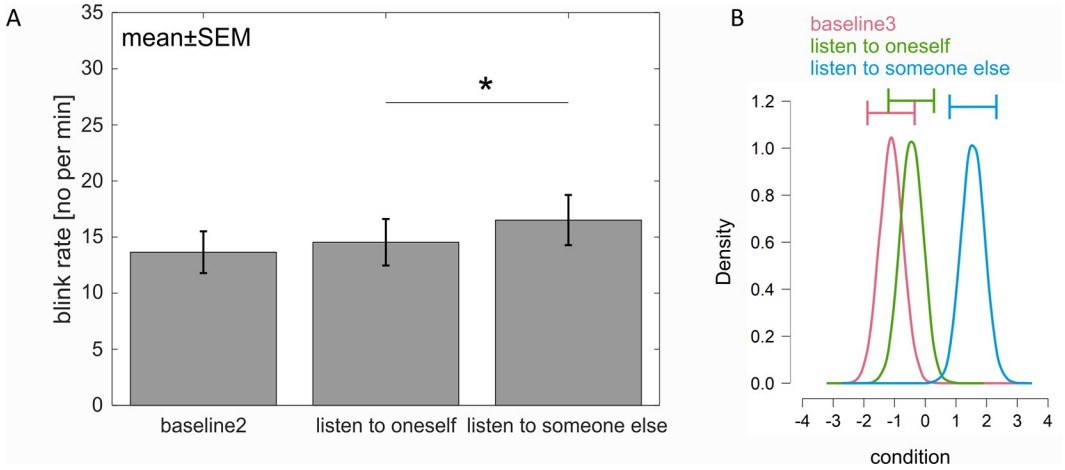

**Fig 4. Influence of auditory input on the blink rate.** A. Blink rate during the rest ("baseline 3"), "listen to someone else" and listening to a previously recorded monologue. Error bars represent one SEM. Stars mark significant differences revealed by parametric statistics. B. Posterior distributions of the effect of each condition on the blink rate. The blink rate between "listening to someone else" was not different to "baseline3", but the blink rate between "listening to oneself" and "listen to someone else" was different. The horizontal error bars above each density represent 95% credible intervals.

without motor output seem to have, if at all, little effect on our blinking. Various researchers have investigated the influence of cognitive load on blink rate, but even the use of similar tasks across different studies, e.g. mental arithmetic, revealed contradictory outcomes. While some researchers showed a negative correlation between blink rate and cognitive load during mental arithmetic [24,25], other studies found an increase in blink rate for difficult arithmetic compared to rest or easy arithmetic [5,23]. The advantage of an arithmetic task, in comparison to our task, namely talking about a given topic, is that one can receive feedback as to the solution for such a task and easily control for the task fulfilment. In experiment 1, we were not able to control for task fulfillment especially during "talking inside the head". We specifically focused on this aspect in experiment 2. However, given the above reviewed work and the contradictory findings, a clear-cut influence of cognition is not indicated.

Similarly, auditory input during listening and self-induced auditory input during talking does not seem to be the cause for the increase in blink rate during a conversation. Listening had no significant effect on blink rate as supported by Bayesian analysis showing that the effect on the blink rate during being at rest ("baseline 3") and during "listen to oneself" is the same. The findings of Bailly, Raidt and Elisei [26] not only fail to show an increase in blink rate due to auditory input, but further suggest an inhibition of blinking during listening periods within a conversation compared to waiting periods. While one is bound to attend to the auditory input of the conversation partner in order to respond accordingly, in our experiment the auditory input was not task relevant. Such a difference in attentional demand might explain the different observations. The differences might also be explained by the fact that our experiment explicitly excluded social interaction. Indeed, it was shown that the duration of blinks can serve as a feedback signal for the conversation partner [8], serving a role in social communication. If social aspects are missing, the reduction of blink rate during listening might also cease. The finding, that "listening to someone else" showed a slightly but significantly increased blink rate compared to "listening to oneself" suggests that the content or characteristics of the auditory stimulation can at least weakly influence blinking.

Our results also indicate that the self-induced auditory input during talking is not the driving factor for the pronounced increase in blink rate, because blinking was significantly enhanced during "talking without sound" (mean: 19.15, SEM: 2.03) which was only slightly less than during "normal talking" (mean: 23.05, SEM: 2.07). Therefore, auditory input as introduced through speaking seems not to exert substantial influences on the blink rate. Importantly, auditory input might alter blink behavior in terms of blink timing. During an attended and continuous stream of auditory input, blinks are seldom elicited shortly before or during stimulus presentation, but rather after stimulus offset [27]. Furthermore, it was shown that blinks are synchronized to the rhythm of auditory presented sentences or even to a specifically attended syllable within a heard sentence [28]. When listening to a monologue, blinks occur predominately at breakpoints of speech or are synchronized with the speaker's blinks [29].

Our findings strongly suggest that motor related factors during talking exert the main influence on the blink rate independent of cognitive or auditory factors. This is indicated by increased blinking during "talking without sound" as well as during the "lollipop" condition. More specifically, by separately investigating the influence of different muscle groups, our results suggest that not all types of motor output are equally linked to blinks. Chewing movements did not significantly increase the blink rate when using a parametric statistical approach, a finding that is in line with previous research [4]. The mouth movements during "lollipop" on the other hand showed a clear effect on blink rate. The prevalent lip movements during "lollipop" and the closeness of motor cortical areas for the lip and eye lid [17] or the innervation of the same nerve [18] might be responsible for this influence. This will be further clarified in experiment 2 and 3.

## Experiment 2

Experiment 1 provides evidence that the motor activity during speaking has a major influence on blinking, while auditory input and cognitive processes only have a minor effect. Our second experiment was designed to replicate the findings of experiment 1, and additionally to describe the underlying causes of the blink rate modulation in greater detail. Concerning cognitive influences, we experimentally manipulated cognitive load by using easy and difficult mental arithmetic tasks and controlled for task fulfillment. In the auditory task, we ensured that participants carefully listen to the spoken words by means of experimental tasks. Concerning motor influences, we isolated defined facial movements, namely lip and jaw movements.

### Method

**Participants.** A power analysis using the effect size of the second analysis of experiment 1 ($n_p^2$ = .263, alpha = .05 and a power of 0.95) suggested a minimum sample size of 22. We tested 23 new participants (mean age: 25.78 years ± 7.60 SD, 6 male) compensating for one potential exclusion. None of the participants took part in experiment 1. All participants gave their written informed consent, agreed to voice recordings and received payment for their participation. The study was conducted in line with the European data protection rules and was approved by the local ethics committee (Institute for Psychology of the Faculty for Human Sciences of the Julius-Maximilians-University of Würzburg; protocol number: GZEK 2020–52).

**Procedure.** Participants sat alone in a moderately lit room. Instructions prior to the task were presented on an Eizo LCD monitor, which was controlled by a Dell Precision M6700 laptop. The monitor turned black during the tasks. The start and end of each trial was marked with a short tone (500 Hz, 100 ms). Binocular eye movements were recorded with 120 Hz using the SMI eye tracking glasses. To record electromyographic (EMG) activity with a sampling rate of 500 Hz, electrodes were placed on the chin, under the left lip corner, on the left

cheek, on the left musculus masseter and above and below the left eye (Fig 1). Two participants lost their chin and lip electrodes during the recording probably due to movement, so we did not attach these electrodes to the last nine participants, in order to prevent them from focusing on the electrodes instead of the task requirements. There was no obvious differences in blink behavior between subjects with four or six electrodes.

The study consisted of nine tasks. Similar to the first experiment, each task was repeated 5 times (except for the baseline, which was repeated 15 times) and lasted for 1 minute each. For an overview of tasks, please refer to Table 2. During "calculating aloud—easy", participants had to count upwards continuously adding one (starting from one) in a normal voice. During "calculating aloud—difficult", they had to continuously subtract seven starting from 200. The same tasks had to be performed in the "calculating inside the head–easy" and "calculating inside the head–difficult", except that they were to perform the arithmetic in their head without moving the mouth and without producing any sound. At the end of these silent trials, participants were asked which number they had reached and how well they performed on a scale from one to seven, where 1 meant "I haven't done the task" and 7 "I was highly concentrated most of the time". During "calculating without sound", participants had to mouth the numbers from one in steps of one without producing any sound. To induce lip movements independent of talking, participants were asked to open and close their lips without moving the jaw ("lip movement"), and to move the jaw up and down without moving the lips during the "jaw movement" task. Again, they scaled their performance from 1 (very bad) to 7 (very good). During the "listen" task, participants had to listen to a voice counting upward (from one in steps of one) leaving out one number that had to be reported after the trial. Again, self-rated concentration had to be indicated on the above-mentioned scale between one to seven. The left out number differed between trials, but was always placed in the second half of the trial. Each analysis included a baseline task where participants were at rest without task. As in experiment 1, the different baseline conditions (1–3) consisted of five randomly, but exclusive, selected minutes out of the 15 minutes. The order of tasks was completely randomized. Participants started each trial by pressing a button at their own pace. The experiment lasted for approximately 65 minutes.

**Blink detection.** We detected blinks based on pupil size as described for experiment 1. In addition, we used the low-passed filtered (20 Hz) data of the electrodes around the eye and detected blinks according to the EOG blink detection described by Wascher [15]. However, we defined the blink on- and offsets as the point where the peak amplitude decreased by three

**Table 2. List of tasks, their description and their use in the analysis of experiment 2.**

| Task | Description | Analysis of which effect |
|---|---|---|
| **"calculating aloud–easy"** | Add 1: 1, 2, 3, . . . | Cognitive (Fig 5) |
| **"calculating aloud–difficult"** | Subtract 7: 200, 193, 186, . . . | Cognitive (Fig 5) |
| **"calculating inside the head–easy"** | Add 1 internally: 1, 2, 3, . . . | Cognitive (Fig 5) |
| **"calculating inside the head–difficult"** | Subtract 7 internally: 200, 193, 186, . . . | Cognitive (Fig 5) |
| **"calculating without sound"** | Mouthing numbers: 1, 2, 3, . . . | Motor (Fig 6) |
| **"lip movement"** | Open and close lips | Motor (Fig 6) |
| **"jaw movement"** | Move jaw up and down | Motor (Fig 6) |
| **"listen"** | Listen to someone adding 1 leaving out one number: 1, 2, . . . 22, 23, 25 . . . | Auditory (Fig 7) |
| **"baseline 1–3"** | Resting | All (Figs 5–7) |

quarters, which slightly differs from the approach used by Wascher and colleagues. For most of the participants, both blink detection methods revealed similar blink numbers, but the eye-tracker data was unusable for three participants and therefore, we present the results based on the EOG blink detection.

**Data analysis.**   We excluded one participant due to a very low blink rate (3.80 blinks/minute) from all analyses. We also excluded trials where participants evaluated their own performance equal or less than 3 on the scale from 1 to 7. This resulted in a list-wise exclusion of two participants from the analysis of cognitive influence on blink rate. One trial of one participant with a blink rate of 110 was also excluded (participant's mean: 27.9 blinks/min). Blink rate during the five minutes of one task were averaged for each participant before the comparisons between conditions.

To evaluate the task demands on performance, we took the entered last number during "calculating inside the head" conditions and extracted the last spoken number that was recorded during "calculating aloud" conditions. This allowed to quantify calculations in the same way for both conditions. If the last number was not a number obtained after correct calculation (only difficult conditions), we counted calculations that were possible up to this point. For example, after 14 calculations, the participant should have arrived at 102, but entered 105. Then, 13 correct calculations were possible and minimally one error. In case of 100, 14 correct calculations were possible. Please note, that this quantification could only result in an overestimation of correct calculations in case of an incorrect last number, and thus, was rather conservative as it made the analysis less likely to find a difference between easy and difficult mental arithmetic. The error identification in the "calculating aloud–difficult" condition, where participants made approximately one error per trial (mean: 1.32, SD: 1.11), supported our approach to only assume one error if the last number was incorrect. As for the blink rate analysis, trials were excluded after which participants evaluated their performance less or equal than 3 on a scale from 1 to 7 (i.e. two participant were excluded for this analysis).

Electromyographical (EMG) data of each electrode was preprocessed by subtracting the mean of all other electrodes in a first step. Subsequently, the data was bandpass filtered between 20 and 90 Hz and a Hilbert transformation was applied. Finally, the resulting EMG amplitudes were averaged over facial electrodes excluding eye-related electrodes.

Implementation and analysis of the experiment was done with MATLAB R2015b (The MathWorks Inc., Natick, MA, USA) in combination with Psychtoolbox [30–32].

## Results Experiment 2

We tested performance with the number of correct calculations based on the last number given by the participants. A repeated measures ANOVA with factors difficulty (easy vs difficult) and condition (calculating aloud vs calculating inside the head) revealed that participants made significantly more calculations during the easy task (add 1) compared to the difficult task (subtract 7) ($F_{(1,19)} = 71.91$, $p < .001$, $\eta_p^2 = .791$) as expected. Moreover, the performance was not significantly better during "calculating inside the head" than during "calculating aloud" ($F_{(1,19)} = 2.89$, $p = .106$, $\eta_p^2 = .132$). Additionally, the interaction was significant ($F_{(1,19)} = 11.83$, $p = .003$, $\eta_p^2 = .384$) showing that participants added more numbers in the "calculating inside the head" condition compared to the "calculating aloud" condition, but made less calculations in the more difficult subtraction task during the "calculating inside the head" condition than during the "calculating aloud" condition (Fig 5A).

Comparable to the analysis of the influence of task demands on performance, the impact of cognitive load on blink rate was examined. To see whether blink rate was increased or decreased during task compared to baseline, we subtracted the blink rate during baseline from the blink rate during the tasks "calculating aloud" and "calculating inside the head" (Fig 5B).

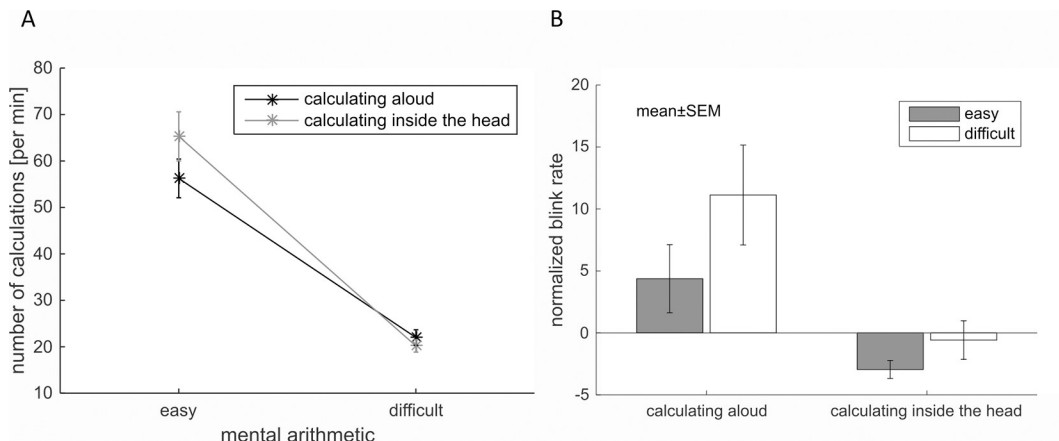

**Fig 5. Results of cognitive tasks.** A. Performance assessment during cognitive tasks. B. Influence of cognitive task demands on the blink rate. Blink rate during "baseline 1" was subtracted from the blink rate in each task showing that only the blink rate in the "calculating aloud–difficult" task was strongly increased. Only the main effect of "calculating aloud" vs "calculating inside the head" was significant ($p < .004$).

While the blink rate increased during "calculating aloud—difficult", the blink rate seemed to be only slightly affected during "calculating inside the head" conditions as well as during "calculating aloud–easy". A repeated measures ANOVA on blink rate with factors aloud/inside the head and easy/difficult revealed a significant increase for aloud tasks compared to quiet tasks ($F(1,19) = 10.43$, $p = .004$, $\eta_p^2 = .354$). In addition, blink rate was higher during difficult tasks compared to easy tasks, but the difference was not significant ($F(1,19) = 2.61$, $p = .123$, $\eta_p^2 = .121$). Also, the interaction was not significant ($F(1,19) = 1.35$, $p = .260$, $\eta_p^2 = .066$).

EMG activity was analyzed in a 1-factor repeated-measures ANOVA across these five tasks ("calculating aloud–easy/difficult", "calculating inside the head–easy/difficult", baseline 1), which revealed a significant difference between the tasks ($F(4,76) = 9.55$, $p < .001$, $\eta_p^2 = .334$, $\varepsilon = .598$ (HF)). Bonferroni-adjusted post-hoc tests revealed increases in EMG activity during "calculating aloud" tasks compared to "calculating inside the head" tasks and all baselines ($ps < .045$) except for the comparison between "calculating aloud—difficult" and "calculating inside the head—easy" ($p = .437$). "Calculating inside the head" tasks did not significantly vary in EMG activity compared to baseline 1 ($ps = 1$) as expected.

Concerning motor tasks, the repeated measures ANOVA comparing the blink rate between motor tasks ("calculating without sound", "lip movement", "jaw movement", "baseline 2") did not reveal a significant effect across tasks ($F(3,63) < 1$) (Fig 6). Neither the "lip movements" nor the "calculating without sound" tasks increased blink rate. The ANOVA comparing EMG activity between these tasks showed a significant main effect ($F(3,63) = 9.73$, $p = .001$, $\eta_p^2 = .317$, $\varepsilon = .525$ (HF)). Bonferroni-adjusted post-hoc tests revealed that the activity during the motor tasks ("lip movement", "jaw movement" and "calculating w/o sound") was significantly increased compared to baseline ($ps < .017$). The EMG activity between the motor tasks did not differ ($ps > .060$).

Finally, participants performed perfectly on the auditory task (100% correct) and always rated their concentration higher or equal to 4 on the 7-point scale (except for two trials). This proves that the cognitive load was quite low and that participants actually listened to the presented numbers. A t-test comparing the blink rate between "listen" and "baseline 3" did not reveal a significant difference ($t(1,21) = 1.53$, $p = .141$, $d = .326$) which replicates the results of

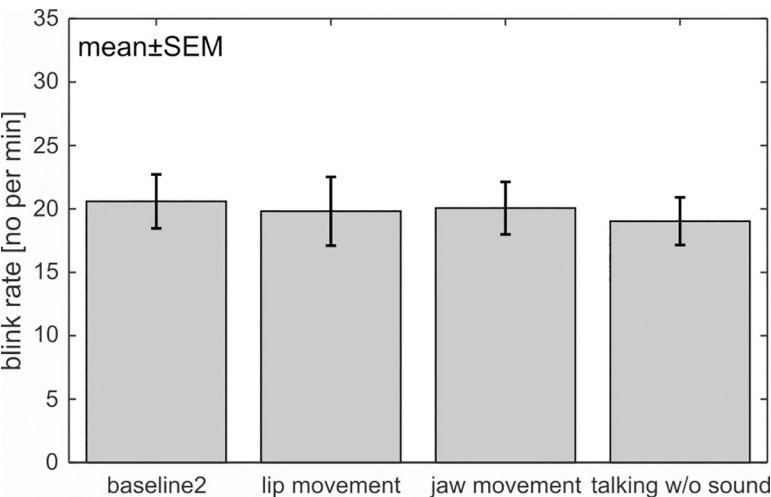

**Fig 6. Influence of motor activity on the blink rate.** Neither isolated "lip movements", nor isolated "jaw movements" influenced the blink rate compared to baseline. During "calculating without sound" participants performed the easy 'Add 1' task, which did not increase the blink rate.

experiment 1 (Fig 7). As expected, EMG activity during these tasks was not significantly different ($t(1,21) = 1.60$, $p = .124$, d = .341).

## Discussion Experiment 2

Our second experiment focused on the cognitive aspects during speaking controlling for task performance. First, trials with low subjective ratings on attentional involvement in the task were excluded. Second, participants had to report the last number of their calculations, which was used to measure performance based on the number of sub-calculations. Participants

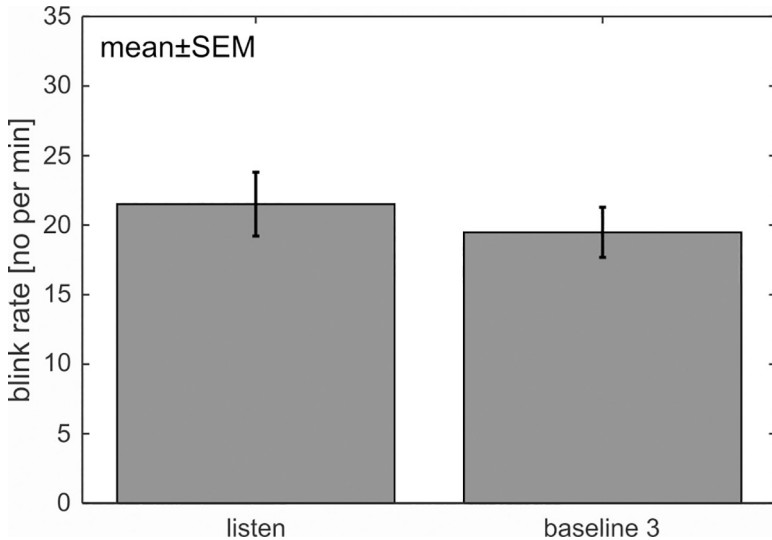

**Fig 7. No influence of auditory input on the blink rate.** Participants listened to the easy 'Add 1' task, where one number was skipped. This number had to be reported afterwards. Blink rate between "listen" and "baseline 3" was not significantly different from each other.

performed significantly better during easy compared to difficult mental arithmetic tasks. This confirms that the addition task was indeed easier than the subtraction task. Importantly, performance was comparable or better during silent conditions compared to normal vocalization showing that participants followed task instructions even during silence. This clearly indicates that the cognitive load was not particularly increased during normal vocalization. After this important step, we did not find a significant difference in blink rate between the easy and difficult task, but there is a tendency towards a higher blink rate during difficult tasks, which has been reported previously [5,23,33]. Our results further showed that an increase above baseline and a clearly visible modulation due to task difficulty was only observed when the task was performed with vocalization. The difference in blink rate between the easy and difficult mental arithmetic when performed in silence was substantially smaller and stayed around baseline level. This small difference of less than five blinks per minute is in line with previous studies using no visual stimulation and no hand movement [5,33]. These findings would suggest that the influence of cognition on blink rate is dependent on additional factors. Indeed, reviewing work on the relation between blink rate and task difficulty shows a rather complex picture. Neither the reading of words compared to the reading of mirror images of the same words [34], nor an easy compared to a difficult letter search task revealed any difference in blink rate [23]. Some other tasks like driving in open country vs in heavy traffic [35] and an easy vs difficult tone counting task [36] show a negative correlation between blink rate and task difficulty. Except for studies investigating conversations [e.g. 4,6,7] or tasks involving spoken responses [22], only few studies show an increase in blink rate during a task compared to rest [37]. In conclusion, whether the blink rate is influence by the cognitive demands of a task seems to be dependent on the specific task requirements. The combined results of experiment 1 showing that "talking inside the head" about an easy topic did not increase the blink rate, and experiment 2 showing that neither easy nor difficult mental arithmetic during silence substantially increased the blink rate compared to baseline, suggest that the cognitive component during a conversation alone is not the driving influence on the blink rate.

Experiment 2 further strengthened the findings of experiment 1 that auditory input during listening does not substantially alter the blink rate. Importantly, this time we controlled if participants indeed attended to the auditory input, because they had to report the number that was left out in the stream of easy calculations. While performance and self-rated concentration on the task was very high, blink rate was not significantly increased. This shows that attending auditory input is not the driving factor for the often reported blink rate increase during conversation.

Experiment 2 additionally broke down speaking into isolated facial movements to concretize our finding that motor output influences blinking. Interestingly, neither isolated lip movements, nor isolated jaw movements increased the blink rate. Consequently, other aspects of speaking modulate blinking behavior. One possibility could be that motor output needs to be combined with a certain amount of cognitive demand. Our results that the blink rate was not substantially affected by "calculating aloud–easy" would point in that direction. However, experiment 1 showed that movements introduced by simply sucking on a lollipop are sufficient to increase the blink rate, which clearly argues against the necessity of cognitive demands. A second possible aspect might be the complexity of movement. Lollipop sucking, as used in the first experiment, does not solely activate isolated lip movements but involves complex muscular activity. Forming full sentences during talking with and without sound as in experiment 1 can also be considered complex motor output [16] and to a certain extent, the utterance of mainly two- and three-digit numbers during the "calculating aloud–difficult" task is possibly more complex than mainly one- and two-digit numbers during the "calculating aloud–easy" task. Unfortunately, to quantify movement complexity, a more sensitive methods,

than the EMG data collection as applied by us would be necessary. A third possibility is a specific involvement of the tongue, as surely can be found during lollipop sucking, but not during isolated lip and jaw movement. The shape and position of the tongue further has a primary function during speech as it shapes the vocal tract [16].

## Experiment 3

In a last experiment, we set out to test a possible involvement of isolated tongue movement on the blink rate increase during speaking and confirm again that motor execution during a complex (cognitive/motor) task leads to an increased blink rate. More specifically, our third experiment compared "normal talking" and "talking without sound" during forming meaningful sentences, with isolated tongue movements and being at rest. In addition to the approach applied in experiment 1, we control for task fulfillment by recording facial EMG activity.

### Method

**Participants.**   24 new participants (mean: 25.00 years, SD: 5.63, 6 male) took part in the third experiment. None of them took part in experiment 1 or 2. The number of participants was chosen upon the power analysis described in experiment 2, which was based on the data of experiment 1 and resulted in 22 participants (+ 2 potential dropouts). All gave their written informed consent and received payment for their participation. The experiment was conducted in line with the European data protection rules.

**Procedure.**   Participants sat alone in a moderately lit room. Auditory instructions were presented via two loudspeakers left and right to the Eyelink 1000 eyetracker (SR Research, ON, Canada). Eye movements were recorded binocularly at a sampling rate of 500Hz. Participants had to touch a horizontally mounted bar with their forehead fixing the distance of the eyes to the eyetracker minimizing large head movements. In addition, electrodes were placed above and below the left eye, under the left lip corner, on the left musculus masseter and below the chin to record the muscular activity of the face and tongue (Fig 1). EMG activity was recorded with 500 Hz. The experiment was controlled by a Dell Precision M6700 laptop.

The study consisted of four tasks (see Table 3). Each task lasted for 1 minute and was repeated five times. As in experiment 1, participants had to talk about easy topics (e.g. "Describe your apartment") during the "normal talking" condition and during the "talking without sound" condition. During the "tongue" condition, participants had to write the numbers from 0 to 9 with their tongue towards the palate in the oral cavity with the mouth closed. Participants had no task and rested during the "baseline" condition. Participants started each trial by pressing a button at their own pace. The trial was preceded and followed by a short auditory tone. The experiment lasted approximately 25 minutes.

**Data analysis.**   Two participants were excluded, because neither the eyetracking data nor the EMG data was usable for blink detection. We used the same EOG and video-based blink detection algorithms as in experiment 2 and the same preprocessing of electromyographical data for muscle activity. The eyetracker data for two other participants showed reduced

**Table 3. List of tasks and their description of experiment 3.**

| Task | Description |
|---|---|
| **"normal talking"** | Talk about a given topic **with** mouth movements and **with** vocalization |
| **"talking without sound"** | Talk about a given topic **with** mouth movements, but **without** vocalization |
| **"tongue"** | Write the numbers from 0–9 with the tip of the tongue |
| **"baseline"** | Resting |

accuracy, which is why we present the results of the EOG blink detection. Please note that the results are similar between the different methods. Again, the implementation and analysis was done with MATLAB R2015b (The MathWorks Inc., Natick, MA, USA).

## Results Experiment 3

Fig 8 shows the blink rate during the four tasks. It was highest for "normal talking" followed by "talking without sound". "Tongue" and "baseline" blink rates were nearly equal and lower than for the other two tasks. A repeated-measures ANOVA comparing the four tasks revealed a significant difference between tasks ($F(3,63) = 24.57$, $p < .001$, $\eta_p^2 = .539$). Bonferroni-adjusted post-hoc tests revealed that every combination is different from the other ($ps < .019$) except for "tongue" vs "baseline" ($p = 1$). EMG activity analysis, taking into account the jaw, lip and tongue electrode, revealed a significant difference between tasks (repeated-measures ANOVA: $F(3,63) = 84.54$, $p < .001$, $\eta_p^2 = .801$, $\varepsilon = .591$ (HF)). Bonferroni-adjusted post-hoc tests showed the expected significant difference between all movements and the baseline ($ps < .003$), no difference between "talking without sound" and "normal talking" ($p = 1$) and a significant difference between "tongue" and the other two movements ($ps < .001$).

## Discussion Experiment 3

Our third experiment replicated the findings of experiment 1 and other studies showing that "normal talking" about a specified topic [4,6,7] increased the blink rate. Moreover, experiment 3 showed again that "talking without sound" requiring similar cognitive effort and motor activity as "normal talking" but lacking auditory components, also significantly increased the blink rate. Adding to experiment 1, this time, task fulfillment was controlled using EMG, which showed that all talking conditions had increased muscle activity compared to baseline. Further, experiment 3 showed that isolated tongue movements were not the driving factor for the increase in blink rate.

Given the finding of experiments 1 and 2 that cognitive demand had only a minor influence on blinking, we assume that complex motor output is the relevant modulator of the blink rate during speaking. The increased complexity of the facial movements during forming sentences compared to counting upwards in experiment 2 ("calculating aloud–easy" and "calculating without sound"), could therefore explain the difference in blink rate modulation between

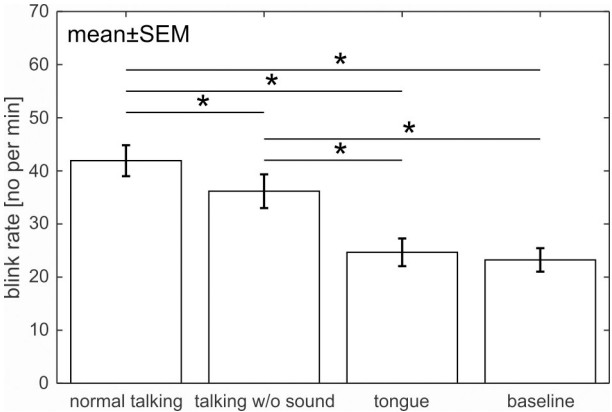

**Fig 8. Influence of motor activity varying in complexity on the blink rate.** All pair-wise post-hoc tests revealed a significant difference in blink rate except for the tasks "tongue" and "baseline". EMG analysis showed that participants fulfilled task requirements.

conditions. Accordingly, the additional motor activity (e.g. of the respiratory system and larynx) during vocalization leading to an increased complexity of motor activity as compared to "talking without sound", could explain the stronger increase in blink rate for "normal talking" compared to "talking without sound".

## General discussion

In sum, we found that neither cognitive demands without verbalization, nor isolated movements of the lips, jaw or tongue, nor the auditory input during vocalization or listening influenced the blink rate. However, our three experiments clearly showed that complex motor tasks as well as verbalization of cognitively demanding tasks increased the blink rate.

During a conversation, we speak at a rate of 3–5 syllables per second [38], which refers to approximately 200 words per minute (language dependent). Given the amount of muscles that are involved in speech production, this motor activity can be described as complex [16]. In our experiments, "normal talking" is the most complex movement followed by "talking without sound", "lollipop", "gum" chewing and finally isolated facial movements. Since we could find blink rates during these tasks in descending order, the complexity of facial motor activity is likely a relevant factor for the amount of blink rate enhancement. An influence of articulation complexity on blinking was touched by von Cramon and Schuri [9] who compared the possibly more complex mouth movements during reciting numbers from 100 upward and the simpler movements during reciting the alphabet. We added a stringent control for auditory and cognitive influences, and excluding these as possible explanations strengthened the evidence that motor activity influences blinking.

Previous research revealed various interactions between different types of movements. For example, blink and saccade rate increases with walking speed [39] and is especially high around the stance phase of the gait cycle [40]. Furthermore, finger tapping entrains spontaneous blinking [41] and a large saccade size holds an increased blink probability [42]. (Micro-) Saccades further co-occur with head movements [43,44] and saccades and reach movements can influence each other's trajectories [45]. This suggests a common phenomenon of motor interaction beyond speaking and blinking. Moreover, our results add to theories on cross-modal multiple action control that demonstrated that eye-related responses are linked to other effector systems such as manual or vocal responses [e.g., 46,47]. Finally, understanding the interaction of movements might advance the realistic visualization of human behavior in artificial avatars thereby possibly improving engagement and/or acceptance of such systems.

Given our results, we advise caution when using blinks as neurological indicators during patient interviews or as indicators of cognitive load during tasks involving verbal responses. In order to obtain optimized blink rate measurements, we suggest to carefully monitor and take into account the duration and complexity of talking, as well as the actual execution of motor output during the evaluation.

## Author Contributions

**Conceptualization:** Mareike Brych, Supriya Murali, Barbara Händel.

**Data curation:** Mareike Brych, Supriya Murali, Barbara Händel.

**Formal analysis:** Mareike Brych, Supriya Murali, Barbara Händel.

**Funding acquisition:** Barbara Händel.

**Investigation:** Mareike Brych, Supriya Murali, Barbara Händel.

**Methodology:** Mareike Brych, Supriya Murali, Barbara Händel.

**Project administration:** Barbara Händel.

**Resources:** Barbara Händel.

**Validation:** Mareike Brych, Barbara Händel.

**Visualization:** Mareike Brych, Barbara Händel.

**Writing – original draft:** Mareike Brych.

**Writing – review & editing:** Mareike Brych, Supriya Murali, Barbara Händel.

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
