## [Decision Letter · Decision Letter 0]

28 May 2020

PONE-D-20-10213

How the Motor Aspect of Speaking Influences the Blink Rate

PLOS ONE

Dear Dr. Brych,

Thank you for submitting your manuscript to PLOS ONE. After careful consideration, we feel that it has merit but does not fully meet PLOS ONE’s publication criteria as it currently stands. Therefore, we invite you to submit a revised version of the manuscript that addresses the points raised during the review process.

I have now received two reviews of the manuscript entitled How the Motor Aspect of Speaking Influences the Blink Rate (PONE-D-20-10213) that you submitted to PLOS ONE. I was fortunate to secure reviews from two experts in the field of blinking and speech, both of whom provided strong insight to the strengths and weaknesses of the manuscript.

I read the original manuscript, and then again with comments provided by the reviewers. As you will see, the reviewers found the manuscript to be well written, concise, and well referenced. The topic is relevant to the broad readership at PLOS One, and you made an admiral attempt at disentangling the contribution of cognitive load, speech, and motor movements on blink rate. I also appreciated the use of dual analyses, both frequentist and Bayesian.

There was however pronounced disagreement between the reviewers regarding the manuscript’s suitability for publication in its current form. Ultimately, I concur with the views of both reviewers. Like Reviewer 2, there is much to like in the presented experiment. It is neat, well thought through, and clear. The findings are important, and I believe should be highlighted to our readers. Yet simultaneously, as noted by Reviewer 1, I had concerns regarding the reliability of the tasks. Satisfying Reviewer 1’s concerns will likely involve the collection of new data in the form of a second experiment. As such, I am rejecting the manuscript in its current form, and I invite you to resubmit in the form of major revisions. If you feel that you cannot resubmit within the allotted timeframe, then you may wish to retract and submit a ‘new’ manuscript. If that is the case, I would ask that you please note this on your submission so that I am asked to handle the ‘new’ version.

A primary concern of Reviewer 1 is that we cannot be sure whether participants engaged in “talking inside the head”. I share this concern, but feel that it extends to several other conditions: “talking without sound” and “listen to someone else”. None of these tasks included a reliability measure. For example, you asked your participants to listen to someone else, but did they? Simple reliability check such as “you’ll be asked a question about the person’s conversation” would go a long way to ensure compliance. For “talking without sound”, this again would be easy to confirm with an analysis of video recording, or if you wanted to get precise, facial EMG. I am not suggesting you deploy fEMG, but this would provide clarity that motor movements between chewing gum and sucking on a lollipop differed in the way that you theorised. Again, a video analysis with a human coder would be the most straightforward solution. For “talking inside the head”, you may wish to consult the imagery literature to identify a suitable reliability task.

I shared reviewer 2’s concerns over the reported blink analysis technique. I am not an expert in eye blink analysis, and will defer to their expertise here, and they seem relatively satisfied. However, it needs to be clarified if this is a novel technique. If so, why was a new eye blink technique required? This is a relatively mature field, and there appear to be more robust eye blink detection methods out there. Why was a dimly lit room used with this optic/vision-based hardware? If this is a novel analysis technique, I would like to see some basic reliability data comparing tracking glass-derived blinks against that of a human coder, or that of a validated technique. This is not a methods paper, and so the sample would not need to be large, but sufficient to give confidence in the technique.

The statistics were a strength of the manuscript; however, several issues should be addressed. First, was a prospective power analysis conducted? If not, the implications need to be discussed. In particular, that several conditions in post-hoc tests approached significance, but may not have due to insufficient power. Second, were assumptions of normality etc. assessed prior to running ANOVAs? Like Reviewer 2, the description of ANOVA structure could be made clearer to the reader. For example, “a two-way repeated measures ANOVA was conducted on DV, with X (2 levels: A, B), and Y (3 levels: C, D, E) as within-subjects factors”. Relatedly, the description of the 8 conditions could be improved; I had difficulty tracking them all. There appeared to be a natural grouping, which you may find helpful for improving clarity: baseline (1), speech (3 levels: inside head, movement without sound, normal), motor (2 levels: lollipop, gum), listening (2 levels: someone else, self-recorded). Please also report the software used to conduct statistical analyses, along with associated packages. Finally, a policy of PLOS One is to make all underlying data available with the manuscript. Please ensure that you adhere to this policy.

We look forward to receiving your revised manuscript.

Kind regards,

Steven R. Livingstone

Academic Editor

PLOS ONE

Journal Requirements:

"granting body: Ethic commission of the institute of Psychology of the Philosophical Faculty II of the Julius-Maximilians-University Würzbu

protocol number: GZEK 2015-01

title of study: Brain and body rhythms – on the relationship between movement and cognition

All participants gave their written informed consent.".   

Please amend your current ethics statement to confirm that your named institutional review board or ethics committee specifically approved this study.

Additional Editor Comments (if provided):

Reviewers' comments:

Reviewer's Responses to Questions

**Comments to the Author**

1. Is the manuscript technically sound, and do the data support the conclusions?

Reviewer #1: Partly

Reviewer #2: Yes

2. Has the statistical analysis been performed appropriately and rigorously? 

Reviewer #1: Yes

Reviewer #2: Yes

3. Have the authors made all data underlying the findings in their manuscript fully available?

Reviewer #1: Yes

Reviewer #2: Yes

4. Is the manuscript presented in an intelligible fashion and written in standard English?

Reviewer #1: Yes

Reviewer #2: Yes

5. Review Comments to the Author

Reviewer #1: The present study examined which factor induced an increase in blink rate during speaking by comparing blink rate between various conditions. They revealed that the motor activity during speaking was a major factor to increase blink rate. They carefully designed the research and the data analysis using Bayesian estimation was performed properly.

The main claim of the present study is that neither the cognitive processes nor the auditory input, but rather, the motor activity of the mouth has the main influence on our blink rate. They conclude this statement depending on the finding that the blink rate during speaking was significantly higher than that during speaking inside the head. However, I cannot agree to their conclusion because it is quite difficult to assess whether participants will seriously generate the sentence during speaking inside the head. There is a high possibility that the cognitive load between those two conditions were different. Large amount of previous studies consistently reported that cognitive load and arousal significantly influence on blink rate. The present study contradicts with those findings. Further experiment is necessary to support the current findings. For example, an experiment examines whether the blink rate changes when the cognitive load is changed due to the difficulty level of the topic under the same conditions for the motor activity of speaking.

Reviewer #2: This is a very interesting, concisely written and important study on the relationship between blink rate and cognitive and motor factors during speech. The idea is very clear: blink rate is increased during speaking, but is this due to the movements involved in speaking or the cognitive effort required of the speaking act. The present study is an elegant one that tries to de-confound the motor and cognitive functions and examines the results in blink rate. I have a fair few comments related to the clarity and completeness of the introduction and methods, but I find the experiment sound, the analysis valid, and the conclusion reasonable. Indeed, I believe this type of work that seeks to test the validity of ‘biomarkers’ of critical importance for the long-term health of science and would like to see the importance of the contribution underlined. To that end, I sincerely you find my comments informative and useful.

1) blink rate (perhaps include a definition) is assumed to be a physiological indicator of cognitive load. I'm not entirely sure what perceptual load could be, perhaps attentional is meant. Some people have suggested that blink rate is a correlate for dopaminergic activity (c.f. Colzato et al., 2008, Neuropsychologia), which might explain the relation with clinical diagnosis. This will also tie neatly into some of the existing controversies in the cognitive load – blink rate correlation as some failures to replicate have been reported (Sescousse et al., 2018, European Journal of Neuroscience; Dang et al., 2017, eNeuro; although Van Slooten et al., 2019, Psychopharmacology).

2) I would suggest slight restructuring of the first paragraph to first explain blink rate, what it does and why it would involve cognitive load and so on, and then to move the correlation between blink rate and speaking to a second paragraph - currently the second sentence in paragraph 1 seems like a strange shift of focus ("Also, the often found increase in blink rate during conversation").

3) Procedure: What does 'no task' mean in the baseline? Is this like a resting state recording in an fMRI experiment?

4) Procedure: Were participants told to come up with topics for themselves or was some randomisation of task x topic done? Perhaps it would help the reader if you could give a table with each of the 8 conditions with a short explanation.

5) Procedure: I imagine participants didn't immediately know what to say during talking conditions, was the length of the task 1 minute from onset of talking or onset of instruction?

6) Data analysis: Digitally recorded speech can be represented as waveforms, it's strictly speaking not transformed. Also, I do not understand what 'which were controlled for outburst signalling continuous talking' means. What is controlled? Is an 'outburst' a sudden high intensity or do you just mean if there's any amplitude in the waveforms it suggested continuous talking? Or is a noise gate filter applied?

7) Blink detection: This is to some extent part of data analysis and might be better presented as part of that.

8) Blink detection: I have some problems with identifying the blink with the pupil radius. That is, I understand that you get pupil radius values from the SMI recordings, but 1) pupil radius tends to be very strongly affected by whatever people are looking at, and 2) if people close their eyes, their pupils increase in size (due to lower light intensity reaching the eye through the lid), not decrease. Perhaps if SMI doesn't see a pupil, or the pupil is partially occluded, it outputs values of them that are small, but that seems an idiosyncracy of the equipment. That said, I find the detection clear enough, it's just that the biological (pupil size, blink) should not be confused for the technological. Please rewrite the preprocessing steps with this in mind.

9) Results: In general, I would advice a little bit of preamble and restructuring. Currently, the analyses seem to drop one after another and it is not always clear what hypotheses you are exactly trying to prove/disprove. Am I correct you are first trying to estimate the effect of talking on blink rate, the basic effect of interest; then to see whether this effect is best approximated by lip movements rather than chewing movements; and then by listening? Both the introductions and method sections could more easily provide predictions, and overview of analysis.

10) Results: The second factor ('repetition') makes very little sense if there is no clear manipulation. If the randomisation was by condition first and repetition second (repeating each condition once before re-randomising) rather than full randomisation, then the repetition suggests a factor of time, which could of course have an effect, but this is not explained.

11) Results, classic analysis: I am not sure which factor is used in the 2-way repeated measures analysis: is it between the 3 conditions, between the 8, or between the 3 and then the 5 remaining? Please use a standard type phrase ('A repeated measures ANOVA with Condition (baseline vs normal talking vs talking inside the head) and Repetition (1st to 5th) as factors - or something to that tune, then write out the analysis. I think the rest of the analysis concerns the differences between the 3 conditions, but what happened with the other 5?

12) Results, Bayesian: A Bayesian analysis is not done to assess the magnitude of differences (that's the effect size), but rather to estimate the evidence of the hypothesis given the evidence. However, I am not entirely sure what the null- and alternative hypothesis are - please specify. It would make things a bit easier if the repetition factor were entirely eliminated from the analysis (as the null-model can both include and not include it currently). Instead, if you are interested in revealing magnitude of differences and the partial eta squares provided are not enough (they rarely are), I would appreciate a simple magnitude (e.g. normal talking increased blink rate by ca. 100%).

13) Discussion: The conclusions seem sound and follow the data analysis. I was a bit disappointed that the authors did not follow up their analysis with an exploratory analysis of the number of syllables uttered and blink rate, as that seems not too prohibitive an amount of additional work. Perhaps that might be added as supplementary analysis?

14) Discussion: I very much agree with the suggestion that blink rate should not be seen as somehow a pure indicator of cognitive load. As to that, I would recommend the authors to phrase their caution even stronger, and not merely in the context of patient interviews, but also experimental work. It is quite possible, for example, that cognitive load or task difficulty could lead to affective responses expresed by the mouth, which may then lead to measurements of blink-rate patterns that are only indirectly related to the suggested causal mechanism (c.f. Maffei & Angrilli, Physiology and behavior, 2019). In that sense, the present study had a very clear focus on speech, but the results may well transfer to other domains, and the finding that blink rate does not provide a process-pure measure of cognition (or emotion, for that matter) is a critical one to get across.

6. PLOS authors have the option to publish the peer review history of their article (what does this mean?). If published, this will include your full peer review and any attached files.

Reviewer #1: No

Reviewer #2: Yes: Michiel Spape

---

## [Author Response · Author response to Decision Letter 0]

28 Apr 2021

Please see our answers in the uploaded file "response to reviewers"

---

## [Decision Letter · Decision Letter 1]

25 Aug 2021

PONE-D-20-10213R1

How the Motor Aspect of Speaking Influences the Blink Rate

PLOS ONE

Dear Dr. Brych,

Thank you for submitting your manuscript to PLOS ONE. After careful consideration, we feel that it has merit but does not fully meet PLOS ONE’s publication criteria as it currently stands. Therefore, we invite you to submit a revised version of the manuscript that addresses the points raised during the review process.

Since the editor for this manuscript was not available at this time to continue handling the paper I have taken over  as active editor for the current revision. As you will see below, the manuscript was seen by a third reviewer who raised several issues. Please address these issues as best as you can. I look forward to receiving your revision.

We look forward to receiving your revised manuscript.

Kind regards,

Markus Lappe

Academic Editor

PLOS ONE

Journal Requirements:

Reviewers' comments:

Reviewer's Responses to Questions

**Comments to the Author**

1. If the authors have adequately addressed your comments raised in a previous round of review and you feel that this manuscript is now acceptable for publication, you may indicate that here to bypass the “Comments to the Author” section, enter your conflict of interest statement in the “Confidential to Editor” section, and submit your "Accept" recommendation.

Reviewer #2: All comments have been addressed

Reviewer #3: (No Response)

2. Is the manuscript technically sound, and do the data support the conclusions?

Reviewer #2: Yes

Reviewer #3: Yes

3. Has the statistical analysis been performed appropriately and rigorously? 

Reviewer #2: Yes

Reviewer #3: Yes

4. Have the authors made all data underlying the findings in their manuscript fully available?

Reviewer #2: Yes

Reviewer #3: No

5. Is the manuscript presented in an intelligible fashion and written in standard English?

Reviewer #2: Yes

Reviewer #3: Yes

6. Review Comments to the Author

Reviewer #2: Thank you for one of the most solid responses to reviewers I have ever had the pleasure to receive. I'm impressed by the changes (including additional experiments), the completeness of the response letter, and the final manuscript. No further comments!

Reviewer #3: Comments

Introduction

The research aim seems to be clearly stated in L64-L66. However, I find it difficult to extract a clear rationale for the research in general. Why is it interesting to find a relationship between blinking behavior and isolated muscle movement and potential concurrent cognitive effort? The authors state in L41-L44: “A clarification of the influence of motor activity seems relevant, especially since blinks serve as neurological indicators in clinical settings. For example, Parkinson’s disease is associated with very low blink rates [8], while high blink rates are observed in patients with Schizophrenia [9].”. It is however not clear to me how the relationships between blinking rate and the different tested conditions could be clinically related to e.g., the aforementioned illnesses. Please provide a clearer rationale. If blink rate is not mediated by dopaminergic hypo- or hyperactivity, and – as suggested by the authors – hypothetically through “facial motor activity”, then the authors should cite plenty sources as to the origins of this hypothesis. See also the next comment.

In L67-L75 the authors continue to provide some hypotheses, however I find no references to literature that give rise to such hypotheses in the first place. Please cite relevant sources that back your claims. Also, specifically in L71-L75 the authors already discuss the results from the first experiment. I believe they have done this they appended the paper with further experiments, However, it is no good practice to discuss results already in the introduction, so please omit this section from the introduction.

Methods exp.1

Please include a visual schematic as to how the subjects were seated relative to the experimental equipment. (if identical for all three experiments, one schematic would suffice)

The eight different experimental conditions, e.g. “talking inside the head” and “lollipop” seem not to be sufficiently motivated. Why have the authors choses specifically these activities and could they cite other sources on the credibility of these conditions.

What does “Additionally, we visually inspected the waveforms representing speech to control for task fulfillment in the “normal talking” condition.” Mean?

Could the authors provide us with a pseudocode of the novel blink detection algorithm? And also discuss how it differs from other state-of-the-art methods?

Results exp.1

The authors should elaborate more on the added value of Bayesian analysis in comparison to the (well executed) regular frequentist statistics.

Discussion exp.1

From L274 onwards, the aim of the second experiment is explained, however it feels more appropriate to include this text in a small introduction section - similar to L29-L75 – rather than putting it in the introduction section of experiment 1.

Experiment 2:

Blink detection: are eye-tracker and EOG data comparable in temporal resolution with respect to blink detection? Could the authors include a metric that quantifies the overlap between the two methods?

It is not clear to me how performance on the mental arithmetic conditions between (and within) participants differed. What was the general level of performance of the participants? Is there any correlation between task performance and blink rate? Also: for me it is not entirely clear how the authors defined an “error” in case of ‘arithmetic in the head’. E.g. if in a minute a subject subtracts 7, 14 times from 200. Ideally the subject would arrive at 102. However, what if the subject reported 100? Or 104? How did the authors infer a performance metric from these ‘singular’ reports?

L485-L491 – same comment as above. It seems that a small, separate introduction seems appropriate

Experiment 3:

Was a prospective power analysis done for experiment 3, if so, could the authors report the results?

L507-L508: was the baseline condition repeated for 15 times, as in the other experiments? Please report on this.

In general: it was not reported if the participants differed between experiments or if the participants were different. This seems important, as I would expect a “lollipop” condition with experiment 3. Why did the authors not include a “lollipop” condition in experiment 3 for comparison purposes? It seems to me that this was the primary reason for doing experiment 3 in the first place.

In L562-L563 the authors start talking about pupil diameter and the usage of short syllables. To what extent is this observation valid within the scope of this paper? The introduction of these subjects seems rather ad hoc, and I would advise to omit the sentence.

General discussion:

L588-L592: it would be nice if the authors could circle back to the general intro, by explaining how the current results are related to the larger picture they paint – the clinical applicability of blinking as a neurological indicator. Since the authors did not include a between subject condition in which they tested several groups, e.g., people with Parkinson’s or schizophrenia, it seems that the clinical applicability of the current results are not as evident as they put it here. As said above, this general (‘clinical’) narrative seems not to be backed by the experiments here, and needs to be revised, both in the general introduction, as well as in the general discussion.

7. PLOS authors have the option to publish the peer review history of their article (what does this mean?). If published, this will include your full peer review and any attached files.

Reviewer #2: No

Reviewer #3: No

---

## [Author Response · Author response to Decision Letter 1]

10 Sep 2021

Our answers to the reviewer's comments can be found in a point-by-point manner in the responseToReviewer_minorRevision.docx file.

---

## [Decision Letter · Decision Letter 2]

27 Sep 2021

How the Motor Aspect of Speaking Influences the Blink Rate

PONE-D-20-10213R2

Dear Dr. Brych,

We’re pleased to inform you that your manuscript has been judged scientifically suitable for publication and will be formally accepted for publication once it meets all outstanding technical requirements.

Kind regards,

Markus Lappe

Academic Editor

PLOS ONE

Additional Editor Comments (optional):

Reviewers' comments:

Reviewer's Responses to Questions

**Comments to the Author**

1. If the authors have adequately addressed your comments raised in a previous round of review and you feel that this manuscript is now acceptable for publication, you may indicate that here to bypass the “Comments to the Author” section, enter your conflict of interest statement in the “Confidential to Editor” section, and submit your "Accept" recommendation.

Reviewer #3: All comments have been addressed

2. Is the manuscript technically sound, and do the data support the conclusions?

Reviewer #3: Yes

3. Has the statistical analysis been performed appropriately and rigorously? 

Reviewer #3: Yes

4. Have the authors made all data underlying the findings in their manuscript fully available?

Reviewer #3: Yes

5. Is the manuscript presented in an intelligible fashion and written in standard English?

Reviewer #3: Yes

6. Review Comments to the Author

Reviewer #3: (No Response)

7. PLOS authors have the option to publish the peer review history of their article (what does this mean?). If published, this will include your full peer review and any attached files.

Reviewer #3: No

---

## [Editor Report · Acceptance letter]

30 Sep 2021

PONE-D-20-10213R2 

How the Motor Aspect of Speaking Influences the Blink Rate 

Dear Dr. Brych:

I'm pleased to inform you that your manuscript has been deemed suitable for publication in PLOS ONE. Congratulations! Your manuscript is now with our production department. 

Kind regards, 

on behalf of

Dr. Markus Lappe 

Academic Editor

PLOS ONE